# Effectiveness of knowledge brokering and recommendation dissemination for influencing healthcare resource allocation decisions: A cluster randomised controlled implementation trial

**Mitchell N. Sarkies**[1,2,3]*, **Lauren M. Robins**[3], **Megan Jepson**[3], **Cylie M. Williams**[3], **Nicholas F. Taylor**[4,5], **Lisa O'Brien**[6], **Jenny Martin**[7], **Anne Bardoel**[8], **Meg E. Morris**[4,9], **Leeanne M. Carey**[10,11], **Anne E. Holland**[12,13], **Katrina M. Long**[3], **Terry P. Haines**[3]

1 Centre for Healthcare Resilience and Implementation Science, Australian Institute of Health Innovation, Faculty of Medicine, Health and Human Sciences, Macquarie University, New South Wales, Australia, 2 Health Economics and Data Analytics Discipline, School of Public Health, Faculty of Health Sciences, Curtin University, Western Australia, Australia, 3 School of Primary and Allied Health Care, Monash University, Victoria, Australia, 4 La Trobe Centre for Sport and Exercise Medicine Research, La Trobe University, Victoria, Australia, 5 Allied Health Clinical Research Office, Eastern Health, Victoria, Australia, 6 Department Occupational Therapy, School of Primary and Allied Health Care, Monash University, Victoria, Australia, 7 Department of Social Work and Human Services, School of Arts, Federation University Australia, Victoria, Australia, 8 Department of Management and Marketing, Swinburne University of Technology, Victoria, Australia, 9 Healthscope Academic and Research Collaborative in Health, Victorian Rehabilitation Centre, Glen Waverly, Victoria, Australia, 10 Occupational Therapy, School of Allied Health, Human Services and Sport, La Trobe University, Victoria, Australia, 11 Neurorehabilitation and Recovery, The Florey Institute of Neuroscience and Mental Health, Melbourne Brain Centre, Victoria, Australia, 12 Department of Allergy, Immunology and Respiratory Medicine, Monash University, Victoria, Australia, 13 Department of Physiotherapy, Alfred Health, Victoria, Australia

* mitchell.sarkies@mq.edu.au

**Data Availability Statement:** Individual participant demographic data cannot be shared publicly

## Abstract

### Background

Implementing evidence into clinical practice is a key focus of healthcare improvements to reduce unwarranted variation. Dissemination of evidence-based recommendations and knowledge brokering have emerged as potential strategies to achieve evidence implementation by influencing resource allocation decisions. The aim of this study was to determine the effectiveness of these two research implementation strategies to facilitate evidence-informed healthcare management decisions for the provision of inpatient weekend allied health services.

### Methods and findings

This multicentre, single-blinded (data collection and analysis), three-group parallel cluster randomised controlled trial with concealed allocation was conducted in Australian and New Zealand hospitals between February 2018 and January 2020. Clustering and randomisation took place at the organisation level where weekend allied health staffing decisions were

because of the potential risk for re-identification. Cluster-level and ward-level data are available within the manuscript and its Supporting information files.

**Funding:** This work was funded by the National Health and Medical Research Council (NHMRC) Australia (APP1114210); https://www.nhmrc.gov.au/ to TPH. The funders had no role in study design, data collection and analysis, decision to publish, or preparation of the manuscript.

**Competing interests:** The authors have declared that no competing interests exist.

made (e.g., network of hospitals or single hospital). Hospital wards were nested within these decision-making structures. Three conditions were compared over a 12-month period: (1) usual practice waitlist control; (2) dissemination of written evidence-based practice recommendations; and (3) access to a webinar-based knowledge broker in addition to the recommendations. The primary outcome was the alignment of weekend allied health provision with practice recommendations at the cluster and ward levels, addressing the adoption, penetration, and fidelity to the recommendations. The secondary outcome was mean hospital length of stay at the ward level. Outcomes were collected at baseline and 12 months later. A total of 45 clusters ($n = 833$ wards) were randomised to either control ($n = 15$), recommendation ($n = 16$), or knowledge broker ($n = 14$) conditions. Four (9%) did not provide follow-up data, and no adverse events were recorded. No significant effect was found with either implementation strategy for the primary outcome at the cluster level (recommendation versus control β 18.11 [95% CI −8,721.81 to 8,758.02] $p = 0.997$; knowledge broker versus control β 1.24 [95% CI −6,992.60 to 6,995.07] $p = 1.000$; recommendation versus knowledge broker β −9.12 [95% CI −3,878.39 to 3,860.16] $p = 0.996$) or ward level (recommendation versus control β 0.01 [95% CI 0.74 to 0.75] $p = 0.983$; knowledge broker versus control β −0.12 [95% CI −0.54 to 0.30] $p = 0.581$; recommendation versus knowledge broker β −0.19 [−1.04 to 0.65] $p = 0.651$). There was no significant effect between strategies for the secondary outcome at ward level (recommendation versus control β 2.19 [95% CI −1.36 to 5.74] $p = 0.219$; knowledge broker versus control β −0.55 [95% CI −1.16 to 0.06] $p = 0.075$; recommendation versus knowledge broker β −3.75 [95% CI −8.33 to 0.82] $p = 0.102$). None of the control or knowledge broker clusters transitioned to partial or full alignment with the recommendations. Three (20%) of the clusters who only received the written recommendations transitioned from nonalignment to partial alignment. Limitations include underpowering at the cluster level sample due to the grouping of multiple geographically distinct hospitals to avoid contamination.

## Conclusions

Owing to a lack of power at the cluster level, this trial was unable to identify a difference between the knowledge broker strategy and dissemination of recommendations compared with usual practice for the promotion of evidence-informed resource allocation to inpatient weekend allied health services. Future research is needed to determine the interactions between different implementation strategies and healthcare contexts when translating evidence into healthcare practice.

## Trial registration

Australian New Zealand Clinical Trials Registry ACTRN12618000029291.

## Author summary

### Why was this study done?

- Healthcare delivery does not always reflect the most up-to-date research evidence.

- There are high levels of evidence to suggest that inpatient allied health services provided during weekends achieve greatest benefits in subacute rehabilitation wards.

- Most weekend allied health services are provided to acute general medical and surgical wards, where there is uncertain evidence of impact.

- Translation of evidence into practice is constrained by a limited understanding of which implementation strategies are most effective for specific settings.

### What did the researchers do and find?

- We conducted a cluster randomised controlled implementation trial to compare the effectiveness of two research implementation strategies across 132 hospitals in Australia and New Zealand.

- We provided hospital managers with either evidence-based weekend allied health practice recommendations or access to a knowledge broker in addition to the recommendations, over a 12-month period.

- Neither implementation strategy was able to be shown effective for ensuring better alignment of weekend allied health provision with practice recommendations; no impacts on hospital length of stay were identified.

### What do the findings mean?

- Evidence dissemination and knowledge brokering are thought to facilitate the translation of research evidence into practice.

- Our study was unable to find whether either of these strategies substantially influenced weekend allied health service decision-making by hospital managers.

- It is possible to study the impact of research implementation using robust trial designs; however, challenges achieving adequate statistical power are a barrier to these evaluations.

## Introduction

Healthcare systems worldwide continue to grapple with unwarranted variation in quality and safety. On average, an estimated 60% of care is delivered according to recommended guidelines [1–4], while up to 30% is considered low-value care [5–7]. Underuse of effective treatments and overuse of those with questionable benefit are arguably responsible for substantial inefficiency and lost opportunity to improve patient outcomes, particularly when considering improvements that could be realised by reallocating resources to high-value care.

The evidence-to-practice gap often manifests through decisions and negotiations within and across multiple levels of the health system [8]. These phenomena create a complex ecosystem, where the delivery of care can be dependent upon policy and managerial decisions

regarding the organisation of resources. One highly topical area of healthcare policy and management decision-making is the provision of weekend allied health services to inpatient wards [9]. Internationally, there is substantial variation in access to allied health professionals within hospitals (e.g., physiotherapists, occupational therapists, speech pathologists dietitians, social workers, and podiatrists) [10–14]. This variability is typified by service provision during weekends, where some hospitals extend limited provision of allied health services [15,16]. A recent systematic review and meta-analysis reported that additional weekend allied health services in subacute rehabilitation units can reduce hospital length of stay by over two days and is a cost-effective approach to improve function and health-related quality of life [17]. However, the benefits were less clear for acute general medical and surgical wards. Despite this evidence, subacute rehabilitation units often provide less allied health during weekends, compared to acute general medical and surgical wards, both in absolute and relative terms [11].

Accessing and applying research evidence to guide resource allocation decisions can be difficult for healthcare managers [18,19]. Therefore, it is imperative to evaluate strategies that promote evidence-informed decision-making to ensure that the benefits of research on weekend allied health service models can be translated to improved health outcomes. Guidelines and recommendations are widely used to disseminate concise instructions regarding patient care. Recommendations have been found to increase awareness of key messages, change attitudes and knowledge [20–22], and impact practice in some circumstances [23–26]. However, they do not always lead to meaningful changes in behaviour [27–30]. Given the feasibility and low cost involved, dissemination of evidence-based practice recommendations may be an efficient way to change practice under certain conditions, even if it is less effective than more interactive alternatives. A more interactive, and resource-intensive, approach is the use of knowledge brokering to support dissemination and implementation of recommendations into evidence-informed decision-making [31–36]. Knowledge brokers are intermediary agents who build relationships between decision-makers and researchers, by sharing expert knowledge and establishing communication channels [37]. Much of this work occurs informally [38]. Yet, these roles are increasingly being formalised and institutionalised [39,40], despite limited evidence to support their effectiveness [36,41]. The substantial cost and resources required to deliver formalised knowledge broker roles [42,43] require evidence of both effectiveness and cost-effectiveness to justify investment. This is because the health system is characterised by finite funding, which must be allocated to competing needs: providing funds for activities with unknown levels of effectiveness such as knowledge brokering is difficult to justify, as those same funds can no longer be used for alternative activities that are known to effectively improve health outcomes elsewhere in the health system.

The aim of this study was to determine the effectiveness of knowledge brokering and dissemination of evidence-based practice recommendations for weekend allied health resource allocation decisions by hospital managers.

## Methods

Monash Health Research Ethics Committee approved this research (HREC/17/MonH/44). The study protocol has been published (S1 Text) [44] and registered (Australian New Zealand Clinical Trials Registry ACTRN12618000029291); Universal Trial Number (UTN): U1111-1205-2621.

### Context

Allied health professionals routinely deliver inpatient services Monday to Friday for hospitals in high-income countries. In certain parts of the world, these services are also extended during

Saturday and Sunday, with Saturday physiotherapy services being the most provided [10,45]. In Australia, approximately 60% of acute hospital wards and 30% of subacute wards provide physiotherapy during weekends [11], which contrasts with Level-I evidence indicating that the benefits are clearer in subacute rehabilitation units [17]. A recent study by Haines and colleagues demonstrated that weekend allied health services could be removed from acute general medical and surgical wards without impacting health or service delivery outcomes and that redesign and reinstatement of these services also did not change these outcomes [16].

## Trial design

In this multicentre study, we conducted a blinded (data collection and analysis) three-group, parallel cluster randomised controlled trial with concealed allocation to compare two alternate research implementation strategies with a control. Clustering occurred at the organisation level where weekend allied health staffing decisions were made within each healthcare organisation to avoid the potential risk of contamination between units of randomisation (e.g., network of hospitals or single hospital). Hospital wards were nested within these decision-making structures. For example, organisations made up of geographically distinct hospitals that made independent decisions in relation to allied health staffing were randomised as separate clusters; those making decisions across hospitals within the organisation were randomised as a single cluster. The diversity in decision-making structures across Australian and New Zealand hospitals constrained the ability to prespecify the number of potentially eligible hospitals/wards to be included within each cluster. Randomisation was stratified based on self-reported geographical classification, as either metropolitan or rural (including regional and remote). This study design is considered the most suitable to address questions of effectiveness, avoid potential contamination across study conditions, and capture outcomes at the system levels where changes were expected to occur.

During the conduct of this study, there was one randomisation error and one modification to the statistical methods described in our study protocol. The randomisation error occurred when one cluster was randomised to the recommendation group but mistakenly did not receive it because of human error. Data were still collected from this cluster and analysed according to the group to which they were assigned. Modifications to the analysis are outlined in the statistical methods section. A CONSORT Extension for Cluster Trials checklist for this study is provided in S2 Text, and the trial was also reported according to the Standards for Reporting Implementation Studies Statement (StaRI) in S3 Text.

## Participants and setting

This study took place across a sample of Australian and New Zealand hospitals. Eligible hospitals were those providing acute or subacute services, with either public or private funding arrangements. Specific wards of interest were general medical and surgical and subacute rehabilitation. Specialist hospitals, including maternity, paediatric, cancer, mental health, and palliative care, were excluded, as no research regarding weekend allied health provision had been identified in these settings. Hospital managers who were responsible for inpatient weekend allied health resource allocation decisions at each cluster were eligible to receive the interventions on behalf of the cluster after providing written informed consent.

## Interventions

A detailed description of the three study conditions according to the Template for Intervention Description and Replication (TIDieR) guidelines [46] is provided in the published protocol [44], and specification of the implementation strategies delivered to the study groups is

**Table 1. Specification and reporting of each implementation strategy.**

| Domain | Recommendation strategy: Written evidence-based practice recommendations | Knowledge broker strategy: Webinar-based knowledge broker in addition to recommendations |
|---|---|---|
| Actor | EviTAH consortium. | EviTAH consortium. Additionally, a single knowledge broker with a PhD-level qualification, from an allied health professional background, with research experience, employed as a postdoctoral research fellow. |
| Action | An evidence-based practice recommendation document provided via email. | An evidence-based practice recommendation document provided via email. Additionally, knowledge broker support for the facilitation, transfer, and exchange of information to enable alignment of practice with the recommendations. Prompting questions informed by the COM-B model [49]. |
| Target of the action | Hospital managers responsible for weekend allied health resource allocation decisions. | Hospital managers responsible for weekend allied health resource allocation decisions. |
| Temporality | Approximately within one week following randomisation. | Approximately within one week following randomisation. |
| Dose | Single occasion (although recommendation resent if requested). | (1) Initial individualised contact made via email or phone to confirm receipt of the written recommendations, discuss local needs, and discuss a plan over the next 12 months; (2) within six months (according to hospital manager availability), a group webinar was arranged; (3) the group webinar was followed up by individualised contact via email or phone (according to hospital manager preference); (4) a final group webinar was arranged; (5) follow up individualised contact thereafter on an "as needs" basis. Contacts were made over a 12-month period with dose varying according to levels of participant engagement. |
| Implementation outcome affected | Primary outcome—practice alignment with recommendations: capturing implementation outcomes—adoption of evidence-based practice recommendation, penetration among eligible hospital wards, and fidelity to the recommendation.<br>Economic, process, and qualitative measures: capturing implementation outcomes—appropriateness of the recommendation as a source of information for the decision, acceptability of the trustworthiness and sufficiency of the recommendation, feasibility of the evidence-base to guide clinical practice, sustainability of the intervention and how it was provided, and cost to make the decision. To be reported in other publications. | Primary outcome—practice alignment with recommendations: capturing implementation outcomes—adoption of evidence-based practice recommendation, penetration among eligible hospital wards, and fidelity to the recommendation.<br>Economic, process and qualitative measures: capturing implementation outcomes—appropriateness of the recommendation as a source of information for the decision, acceptability of the trustworthiness and sufficiency of the recommendation, feasibility of the evidence-base to guide clinical practice, sustainability of the intervention and how it was provided, and cost to make the decision. To be reported in other publications |
| Justification | Evidence-based practice recommendation documents are one of the few implementation strategies that have been evaluated for hospital managers [41,50,51], which have the potential to increase engagement with research implementation [52]. | Multifaceted and interactive implementation strategies are thought to improve evidence-informed decision-making, particularly for organisations without a strong research culture [36]. Many public health organisations have adopted knowledge broker roles [53]. |

COM-B, capability, opportunity, motivation, and behaviour; EviTAH, The Evidence Translation in Allied Health; PhD, post-honorary doctorate.

reported in Table 1 [47]. Each strategy was commenced at the time of randomisation for a period of 12 months to implement specific recommendations regarding weekend allied health provision, derived from a systematic review and meta-analysis [17] and summarised in Box 1. The full evidence-based practice recommendations are provided (S4 Text).

Briefly, three study conditions were delivered at the cluster level: (1) usual practice wait list control; (2) dissemination of written evidence-based practice recommendations; and (3) access to a webinar-based knowledge broker in addition to the recommendations. The waitlist control group experienced usual practice conditions according to their local setting. Upon study completion, they received the evidence-based recommendations for weekend allied health provision. Participants in the recommendation group were provided with an evidence-based weekend allied health practice recommendation (detailed and summarised) document via email. This document contained specific recommendations for the proportion of total allied health services that should be delivered during weekends. The document was constructed with an outline of key messages, executive summary, and presentation of the full research methods

> ## Box 1. Summary of evidence-based policy recommendations for weekend allied health provision
>
> 1. Reduce weekend allied health staffing to a criterion of clinical priorities and exceptions that ensure between 0% and 0.1% of total allied health service events on acute general medical and surgical wards are delivered on weekends.
>
> 2. Increase weekend allied health staffing for subacute rehabilitation units to provide physiotherapy or a combination of physiotherapy and occupational therapy on Saturdays and physiotherapy on a case-by-case basis for stroke patients and, occasionally, to other patients when a clear need is evident on Sundays. This would ensure that between 10% and 20% of total service events for these professions are provided on weekends.

and findings [48]. The knowledge broker group was provided the same recommendation document and additional access to a knowledge broker who facilitated the transfer of relevant information to promote evidence-informed decision-making. The knowledge broker offered support via interactive online webinar, telephone, or email, in one-on-one and group settings. Prompting questions by the knowledge broker were informed by the COM-B (capability, opportunity, motivation, and behaviour) behaviour change model [49]. This support included individual needs assessments and developing a 12-month plan to address the recommendations. These supports followed an iterative process depending on participant needs, based on factors perceived to be associated with effective strategies from a recent systematic review [41]. The dosage and duration of the interventions were based on a similar study evaluating a knowledge broker role [36], which aligned with our hypothesis that a 12-month intervention duration would prove sufficient time for hospital managers to develop and implement a business case for change.

### Outcomes

The primary outcome of interest was whether weekend allied health service provision at both the cluster and ward level aligned with the recommendations at 12-month follow-up. This outcome addressed the adoption of evidence-based practice recommendation, penetration among eligible hospital wards, and fidelity to the recommendation. Alignment with the recommendations was determined according to the number of allied health service events occurring during weekends, as a proportion of the total allied health service events for the cluster or ward, over a one-month period. A ratio of allied health full-time equivalent staffing was used, where service event data were not available ($n$ = 3 clusters). Allied health service event data were collected at the time of randomisation for the preceding calendar month and the same calendar month, 12 months later. For cluster-level analysis, each cluster received a single classification as either (1) fully aligned with the recommendations for both acute wards and subacute units; (2) partially aligned with the recommendations (if acute wards are aligned but subacute units are not aligned, or vice versa); or (3) not aligned with policy recommendations. For the ward-level analysis, each acute ward or subacute unit received a single classification as either (1) fully aligned with the recommendations; or (2) not aligned with policy recommendations.

The secondary outcome was the mean hospital length of stay at each ward for the calendar month 12 months after study entry. We also collected equivalent data for each ward from the same calendar month 12 months earlier. Hospital length of stay was extracted from administrative data sources [54]. Detailed process and economic outcomes described in our protocol are planned for other publications.

## Randomisation

Study investigators consulted with each healthcare organisation to determine their decision-making structure for allied health staffing to reduce the risk of contamination between study groups. Healthcare organisations that made allied health staffing decisions across multiple hospitals were treated as a single unit of recruitment and randomisation, where hospitals with independent decision-making processes within a broader healthcare organisation were treated as separate units. Hospital wards were nested within clusters and randomised using a random number sequence, generated by a single investigator (TPH) using an online software application [55] with permuted blocks within randomisation strata of sizes of 3, 6, or 9. Investigators conducting recruitment, data collection, and analysis (MS and MJ) were blinded, as per procedures outlined in the published protocol [44].

## Statistical methods

Initially, the use of the nonparametric rank-based classical hypothesis test was planned to compare data for the primary outcome. However, changes were made to account for the observation that several clusters and wards were already aligned with the evidence-based recommendations. We changed our analysis approach to an ordered logit regression at the cluster level and logistic regression analysis at the ward level, so we could statistically adjust for the baseline status of each unit of analysis when comparing 12-month follow-up alignment with the recommendations between groups. These ANCOVA style analyses are aligned with recommendations from the Cochrane Handbook for Systematic Reviews of Interventions [56] for how baseline values can be appropriately incorporated into an interventional analysis framework. We used robust variance estimates to account for dependency of clustering across wards that had the same decision-makers, such that this analysis was still analysing data relative to the level of randomisation. The mean hospital length of stay was compared between groups in a ward-level analysis. Linear regression using baseline mean hospital length of stay as a covariate and robust variance estimates at the level of the decision maker was conducted.

Analysis of cluster-level and ward-level data was undertaken according to the group to which they were assigned by an analyst (MS) blinded to group allocation, using three mock codes representing different sequence allocation patterns. Multiple imputation using chained equations was performed to impute 50 datasets for missing values in both baseline and follow-up alignment with the recommendations. Geographical classification was used as the independent variable when imputing missing data at the cluster level; geographical classification and ward classification were used at the ward level. Two sensitivity analyses were conducted: The first was a complete case analysis for comparison with the analysis of imputed data, and the second was a multivariable analysis to adjust for other potential baseline confounders (cluster level: geography and full-time research staffing; ward level: geography and ward type). All analyses were adjusted for the number of weekday days and weekend days within the month to ensure that calendar years changes in the proportion of day types did not impact the results. Analyses were undertaken using Stata version 13.1 (StataCorp, College Station, Texas). Sample size estimates are reported in our study protocol [44].

We planned two levels of analysis, one to be conducted at the cluster level (one unit of data per cluster) and the other to be conducted at the ward level (nested within cluster). The study sample size was determined based on the most conservative unit of assessment for our primary outcome at the cluster level (organisation-level where weekend allied health staffing decisions were made), as an adequate sample size at the cluster level would also prove sufficient at the ward level. Further power analysis at the ward level was not conducted because there was no reasonable way to estimate the likely intracluster correlation coefficient (ICC) for this health-care context and our main sample size constraints pertained to practicality of recruitment at the cluster level. A sample size of 25 clusters per group was estimated to provide greater than 80% power, assuming that 50% of clusters in either intervention group and 10% in the control group completely aligned with the policy recommendations. The diversity in decision-making structures across Australian and New Zealand hospitals constrained our ability to prespecify the number of hospitals/wards within each cluster and recruit sufficient clusters per group, as often allied health staffing decisions were made across hospitals/wards within organisations resulting in fewer potential units of randomisation. Study data used in these analyses are provided (S1 and S2 Data).

## Results

The first cluster was recruited on 7 February 2018, and the final cluster completed their intervention in January 2020; final data collected for that period on 30 April 2020. A total of 45 clusters were randomised to either the control ($n = 15$), recommendation ($n = 16$), or knowledge broker ($n = 14$) groups. There was one cluster that did not provide any outcome data and three that did not provide follow-up outcome data (recommendation $n = 2$; knowledge broker $n = 2$) leaving a rate of 9% loss to follow-up (Fig 1). No adverse events were recorded. Within the clusters, there were $n = 132$ hospitals, $n = 833$ wards, and $n = 204$ hospital managers, whose baseline characteristics are provided in Tables 2 and 3.

The summative raw data for the primary and secondary outcomes are presented in Tables 4 and 5. Proportion of total allied health service events provided on weekends is presented for each group, along with the corresponding number and percentage of clusters or wards aligned with the recommendations. None of the clusters in the control or knowledge broker groups transitioned from "not aligned" at baseline to "partial" or "full alignment" at 12-month follow-up. Three clusters (21%) that did not align to the practice recommendations, from the written recommendation document only group, transitioned to partial alignment within the study period. The flow of cluster alignment from baseline to follow-up, by implementation strategy group, is presented in Figs 2–4 [57]. The most common reasons for missing follow-up data across all three groups were (1) that a different contact person was used (due to staff turnover) who did not know how to extract all the required data; and (2) the contact person did not have time to extract all the required data.

The effect size estimates for the implementation strategies compared to the control are presented in Table 6. Adjusted for baseline recommendation alignment, there was no significant difference for the primary outcome of alignment with the recommendations between the groups at the cluster level using ordered logit regression (recommendation versus control β 18.11 [95% CI −8,721.81 to 8,758.02] $p = 0.997$; knowledge broker versus control β 1.24 [95% CI −6,992.60 to 6,995.07] $p = 1.000$; recommendation versus knowledge broker β −9.12 [95% CI −3,878.39 to 3,860.16] $p = 0.996$). These findings were similar with a multivariable sensitivity analysis that included potential confounding baseline variables: geographic location and whether a full-time allied health researcher was employed (recommendation versus control β 19.65 [95% CI −18,116.13 to 18,155.43] $p = 0.998$; knowledge broker versus control β 1.93

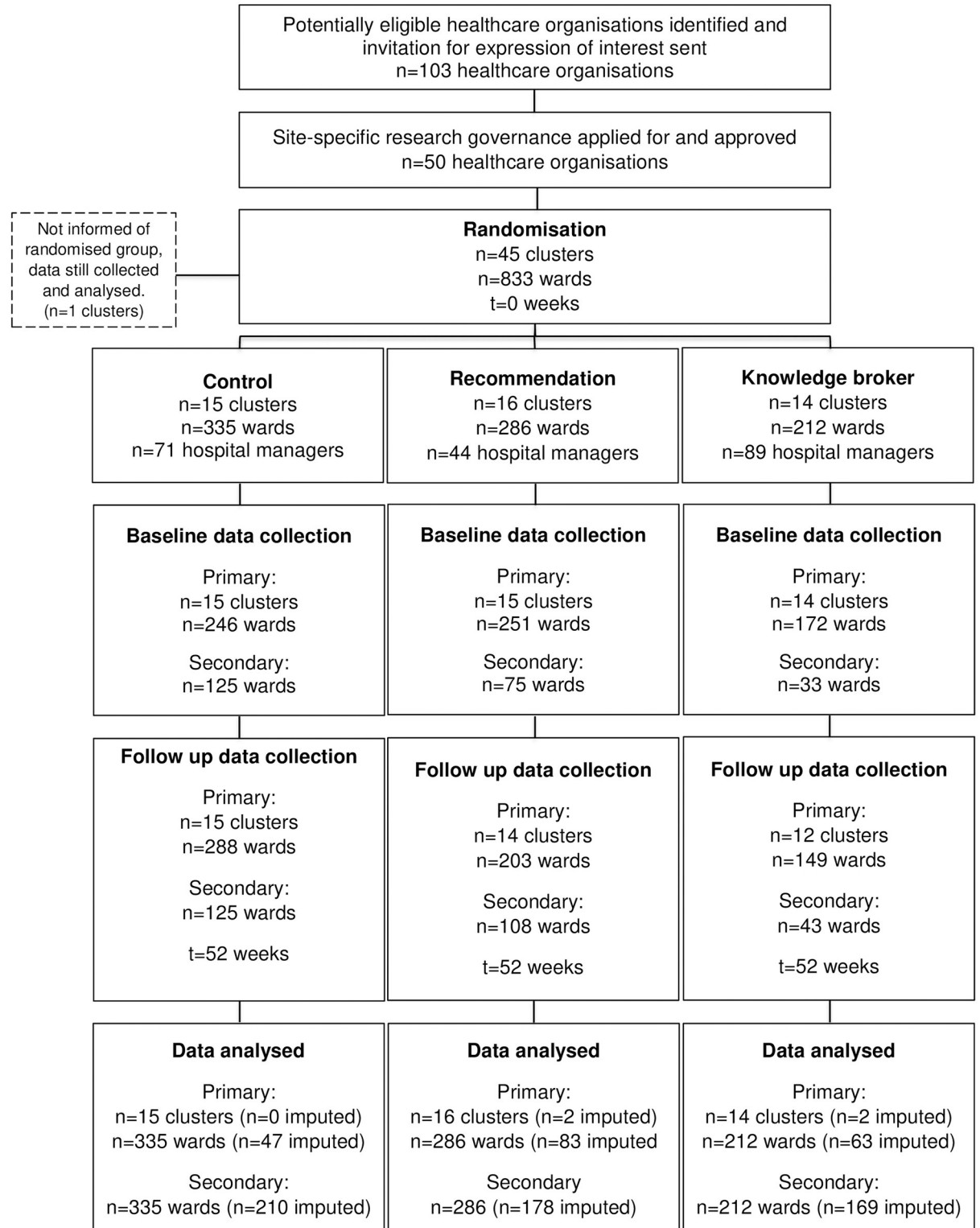

**Fig 1. CONSORT study flow diagram.** One cluster did not provide baseline or follow-up data; three clusters did not provide follow-up data only.

**Table 2. Hospital manager baseline demographics.**

| | Control n = 71 | Recommendation n = 44 | Knowledge broker n = 89 | Total n = 204 |
|---|---|---|---|---|
| Geographical classification n (%) | | | | |
| Metro | 44 (62) | 27 (61) | 51 (57) | 122 (60) |
| Rural | 25 (35) | 17 (39) | 36 (40) | 78 (38) |
| Mix | 2 (3) | 0 (0) | 1 (1) | 3 (1) |
| Unknown | 0 (0) | 0 (0) | 1 (1) | 1 (<1) |
| Hospital classification n (%) | | | | |
| Acute | 63 (89) | 37 (84) | 75 (84) | 175 (86) |
| Subacute | 7 (10) | 4 (6) | 8 (9) | 19 (9) |
| Mix | 1 (1) | 3 (4) | 5 (6) | 9 (4) |
| Missing | 0 (0) | 0 (0) | 1 (1) | 1 (0.5) |
| Age (years) mean (SD) | 46 (8.8) | 47 (9.6) | 46 (9.1) | 46 (9.1) |
| Sex (female) n (%) | 53 (75) | 35 (80) | 71 (79) | 159 (78) |
| Professional background n (%) | | | | |
| Physiotherapy | 22 (31) | 13 (30) | 18 (20) | 53 (26) |
| Occupational therapy | 10 (14) | 11 (25) | 16 (18) | 37 (18) |
| Social work | 12 (17) | 4 (9) | 13 (15) | 29 (14) |
| Dietetics | 10 (14) | 6 (14) | 12 (13) | 28 (14) |
| Speech pathology | 11 (15) | 6 (14) | 13 (15) | 30 (15) |
| Podiatry | 2 (3) | 2 (5) | 6 (7) | 10 (5) |
| Other | 4 (6) | 2 (5) | 11 (12) | 17 (8) |
| Healthcare policy or management experience (years) mean (SD) | 12 (7.6) | 14 (10.6) | 12 (8.9) | 12.4 (8.8) |
| Highest qualification n (%) | | | | |
| Diploma | 1 (1) | 0 (0) | 0 (0) | 1 (<1) |
| Bachelor | 24 (34) | 9 (20) | 35 (39) | 68 (33) |
| Graduate or Honours | 14 (20) | 14 (32) | 24 (27) | 52 (25) |
| Master | 30 (42) | 19 (43) | 29 (33) | 78 (38) |
| Doctorate | 2 (3) | 2 (5) | 1 (1) | 5 (2) |

Percent (%) values subject to rounding error and refer to group totals.

n, sample; SD, standard deviation.

[95% CI −22,189.52 to 22,193.38] $p$ = 1.000; recommendation versus knowledge broker β −9.28 [95% CI −4,601.58 to 4,583.01] $p$ = 0.997). There was no significant difference for the primary outcome of alignment with the recommendations between the groups at the ward level using logistic regression (recommendation versus control β 0.01 [95% CI 0.74 to 0.75] $p$ = 0.983; knowledge broker versus control β −0.12 [95% CI −0.54 to 0.30] $p$ = 0. 581; recommendation versus knowledge broker β −0.19 [−1.04 to 0.65] $p$ = 0.651). These findings were similar with a multivariable sensitivity analysis that included potential confounding baseline variables: geographic location and ward type (recommendation versus control β 0.42 [95% CI −0.62 to 1.46] $p$ = 0.430; knowledge broker versus control β 0.15 [95% CI −0.47 to 0.76] $p$ = 0.639; recommendation versus knowledge broker β 0.04 [95% CI −0.96 to 0.87] $p$ = 0.925).

For the secondary outcome of hospital length of stay, there was no significant difference between the groups at the ward level using linear regression (recommendation versus control β 2.19 [95% CI −1.36 to 5.74] $p$ = 0.219; knowledge broker versus control β −0.55 [95% CI −1.16 to 0.06] $p$ = 0.075; recommendation versus knowledge broker β −3.75 [95% CI −8.33 to 0.82] $p$ = 0.102. These findings were similar with a multivariable sensitivity analysis that included potential confounding baseline variables: geographic location and ward type

**Table 3. Healthcare organisation baseline demographics.**

| | Control | Recommendation | Knowledge broker | Total |
|---|---|---|---|---|
| Clusters (hospital or hospital network) n (%) | | | | |
| Total | **15** | **16** | **14** | **45** |
| Metropolitan | 9 (60) | 9 (56) | 8 (57) | 26 (58) |
| Rural | 6 (13) | 7 (44) | 6 (43) | 19 (42) |
| Allied health research staffing n (%) | | | | |
| Full-time academic | 6 (40) | 5 (31) | 2 (14) | 13 (29) |
| Clinician-researcher | 9 (60) | 7 (44) | 6 (43) | 22 (49) |
| Number of clusters providing inpatient allied health services n (%) | | | | |
| Total weekend | 15 (100) | 15 (94) | 9 (64) | 39 (87) |
| Acute | 15 (100) | 15 (94) | 9 (64) | 39 (87) |
| Subacute | 7 (47) | 10 (63) | 5 (36) | 22 (49) |
| Hospitals n (%) | | | | |
| Total | **45** | **48** | **39** | **132** |
| Metropolitan | 20 (44) | 26 (54) | 18 (46) | 64 (48) |
| Rural | 25 (56) | 22 (46) | 21 (54) | 68 (52) |
| Ward classification n(%) | | | | |
| Total | **335** | **285** | **212** | **833** |
| Acute | 288 (86) | 227 (79) | 175 (83) | 690 (83) |
| Subacute | 47 (14) | 59 (21) | 37 (17) | 143 (17) |
| Ward type n (%) | | | | |
| General medical and surgical | 250 (75) | 191 (67) | 141 (67) | 582 (70) |
| Orthopaedic | 22 (7) | 16 (5.6) | 10 (5) | 48 (6) |
| Neurological | 16 (5) | 15 (5) | 13 (6) | 44 (5) |
| Rehabilitation | 46 (14) | 56 (20) | 30 (14) | 132 (16) |
| Mixed | 1 ($<$1) | 7 (2) | 18 (8) | 26 (3) |

Percent (%) values calculated relative to total values per group and are subject to rounding error.

n, sample.

(recommendation versus control β 1.92 [95% CI −1.85 to 5.70] $p = 0.308$; recommendation versus knowledge broker β −5.32 [95% CI −13.95 to 3.31] $p = 0.213$), although a significant difference was identified for the knowledge broker versus control (β −0.71 [95% CI −1.38 to −0.05] $p = 0.037$).

The primary and secondary outcomes, at the cluster or ward level, were largely unaffected by a complete case sensitivity analysis presented in Table 7.

## Process outcomes

The knowledge broker strategy dose varied from that specified in the protocol across sites, due to differing levels of participant engagement. There were 17 interactive online knowledge broker support webinars (duration one to two hours) conducted in total to facilitate the transfer of relevant information to promote evidence-informed decision-making. Healthcare decision maker participation was higher in the initial (range 6 to 18) compared to follow-up (range 2 to 8) webinars. Typically, the first webinars were attended by representatives from each allied health profession; however, the second webinar was predominantly attended by decision-makers from the physiotherapy and occupational therapy professions because the evidence and recommendations mostly pertained to these roles. Only two clusters (14%) participated in more than two webinars with the knowledge broker. A desire to make an internal decision

**Table 4. Cluster-level summative raw data for primary outcome.**

| | Control (n = 15) | | Recommendation (n = 16) | | Knowledge broker (n = 14) | |
|---|---|---|---|---|---|---|
| | Baseline | Follow-up | Baseline | Follow-up | Baseline | Follow-up |
| Cluster-level allied health service events, per day and ward mean (SD), obs | | | | | | |
| Acute ratio | 0.13 (0.06), 15 | 0.14 (0.07), 15 | 0.13 (0.08), 15 | 0.11 (0.09), 14 | 0.09 (0.14), 14 | 0.08 (0.09), 12 |
| Subacute ratio | 0.10 (0.26), 15 | 0.04 (0.08), 15 | 0.04 (0.08), 15 | 0.06 (0.13), 14 | 0.06 (0.13), 14 | 0.04 (0.07), 12 |
| Cluster-level alignment with recommendations n (%) | | | | | | |
| Full alignment | 0 (0) | 0 (0) | 0 (0) | 0 (0) | 0 (0) | 0 (0) |
| Partial alignment | 0 (0) | 0 (0) | 1 (6) | 3 (19) | 8 (57) | 5 (36) |
| Not aligned | 15 (100) | 15 (100) | 14 (88) | 11 (69) | 6 (43) | 7 (50) |
| Missing | 0 (0) | 0 (0) | 1 (6) | 2 (13) | 0 (0) | 2 (14) |

Percent (%) values calculated relative to total values per group and are subject to rounding error; acute and subacute ratios were calculated as the number of allied health service events occurring during weekends, as a proportion of the total allied health service events for the cluster, over a one-month period; acute ratio alignment with the recommendation is between 0 and 0.001; subacute ratio alignment with the recommendation is between 0.1 and 0.2; missing values removed from analysis.

n, sample; obs, observations; SD, standard deviation.

**Table 5. Ward-level summative raw data for primary and secondary outcomes.**

| | Control (n = 335) | | | Recommendation (n = 286) | | | Knowledge broker (n = 212) | | |
|---|---|---|---|---|---|---|---|---|---|
| | Baseline | | Follow-up | Baseline | | Follow-up | Baseline | | Follow-up |
| Ward-level allied health service events, per day and ward mean (SD), obs | | | | | | | | | |
| Acute ratio | 0.12 (0.15), 214 | | 0.11 (0.15), 246 | 0.11 (0.18), 199 | | 0.10 (0.17), 155 | 0.13 (0.18), 147 | | 0.09 (0.12), 123 |
| Subacute ratio | 0.06 (0.10), 32 | | 0.05 (0.08), 43 | 0.06 (0.17), 52 | | 0.03 (0.08), 48 | 0.09 (0.16), 25 | | 0.08 (0.20), 26 |
| Ward-level alignment with recommendations n (%) | | | | | | | | | |
| Acute aligned | 70 (21) | | 72 (21) | 67 (23) | | 51 (18) | 63 (30) | | 47 (22) |
| Acute not aligned | 144 (43) | | 173 (52) | 132 (46) | | 104 (36) | 84 (40) | | 76 (36) |
| Acute missing | 74 (26) | | 43 (15) | 28 (12) | | 72 (32) | 28 (16) | | 52 (30) |
| Subacute aligned | 3 (1) | | 7 (2) | 1 (<1) | | 2 (<1) | 3 (1) | | 4 (2) |
| Subacute not aligned | 29 (9) | | 36 (11) | 51 (18) | | 46 (16) | 22 (10) | | 22 (10) |
| Subacute missing | 15 (32) | | 4 (11) | 7 (12) | | 11 (19) | 12 (32) | | 11 (30) |
| Ward-level hospital length of stay mean (SD), obs | * | † | | * | † | | * | † | |
| Acute | 7.1 (9.1), 111 | 6.77 (7.5), 101 | 7.4 (7.4), 108 | 12.6 (13.1), 81 | 12.5 (13.3), 76 | 13.2 (18.0), 82 | 9.8 (7.8), 45 | 5.6 (4.4), 25 | 5.1 (3.3), 40 |
| Missing n (%) | 177 (61) | 187 (65) | 180 (63) | 146 (64) | 151 (67) | 145 (64) | 130 (74) | 150 (86) | 135 (77) |
| Subacute | 19.3 (15.1), 14 | 19.3 (15.1), 14 | 24.1 (15.0), 16 | 27.4 (25.1), 27 | 28 (26.0), 25 | 26.6 (21.7), 25 | 20.6 (13.6), 8 | 11.6 (5.9), 5 | 14.3 (6.5), 6 |
| Missing n (%) | 33 (70) | 33 (70) | 31 (66) | 32 (54) | 34 (58) | 34 (58) | 29 (78) | 32 (86) | 31 (84) |

Percent (%) values calculated relative to total values per group and are subject to rounding error; acute and subacute ratios were calculated as the number of allied health service events occurring during weekends, as a proportion of the total allied health service events for the cluster, over a one-month period; acute ratio alignment with the recommendation is between 0 and 0.001; subacute ratio alignment with the recommendation is between 0.1 and 0.2; missing values removed from analysis.

n, sample; obs, observations; SD, standard deviation.

*Includes all baseline wards where data provided.

†Only includes baseline wards where follow-up data also provided.

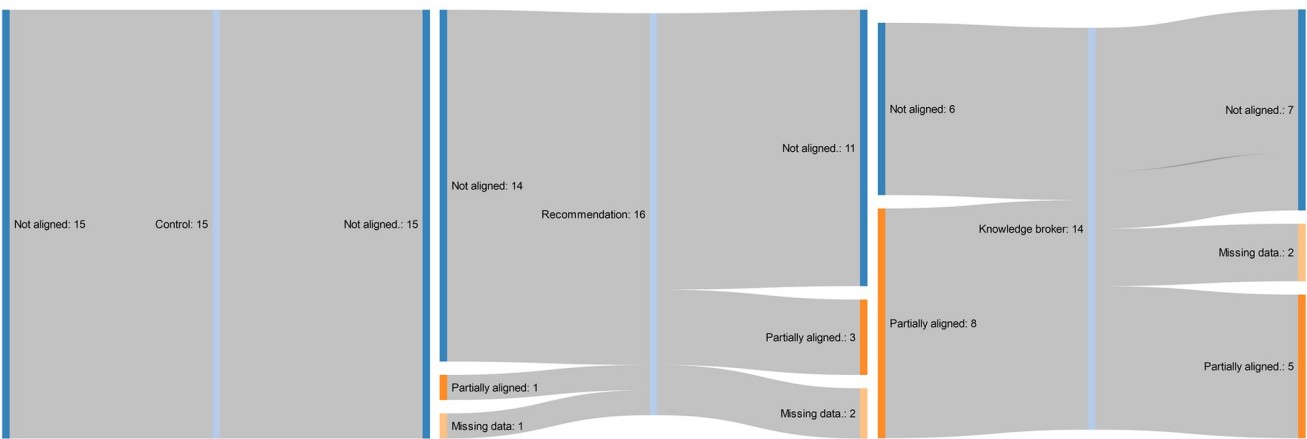

**Fig 2. Flow of clusters from baseline to follow-up policy recommendations alignment, by implementation strategy group.**

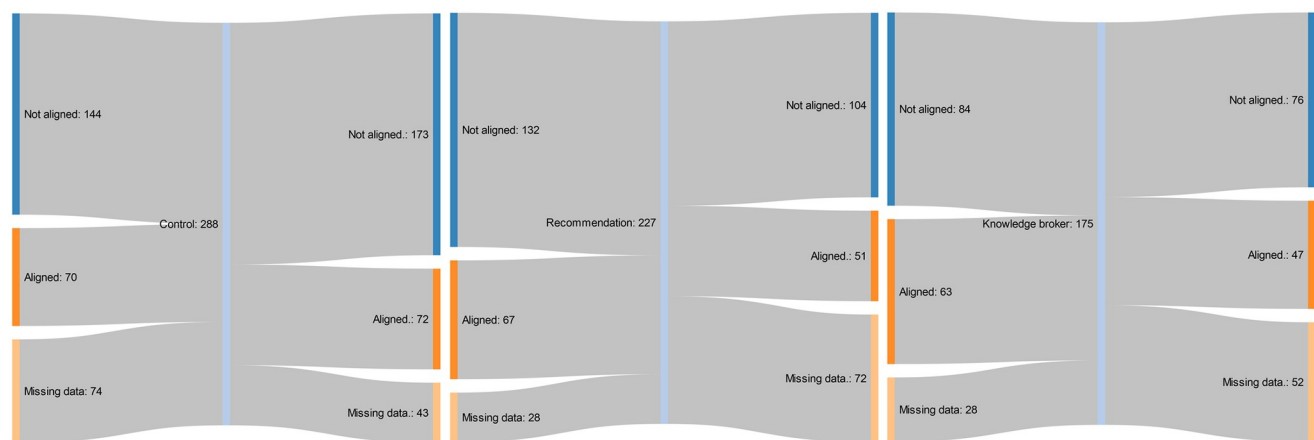

**Fig 3. Flow of acute wards from baseline to follow-up policy recommendations alignment, by implementation strategy group.**

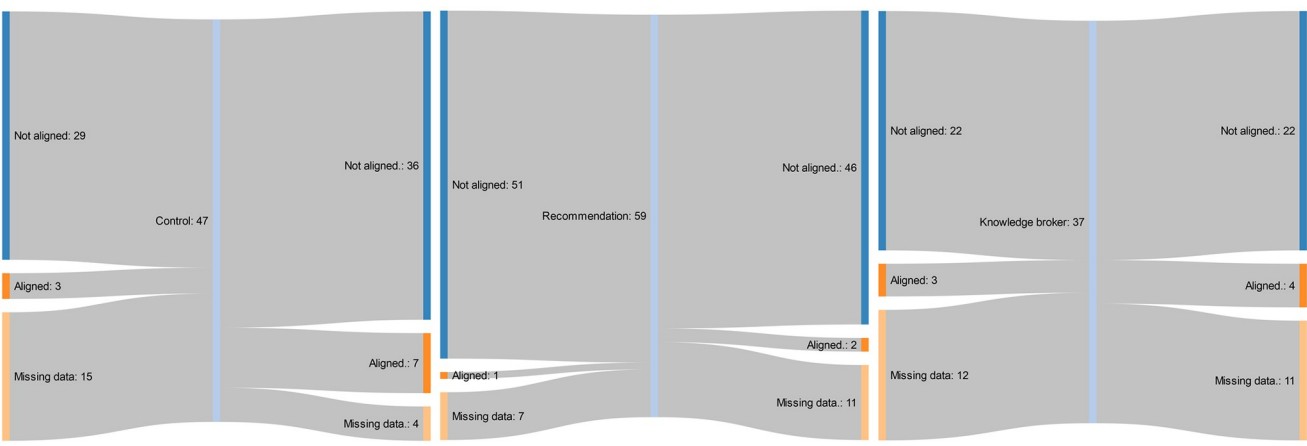

**Fig 4. Flow of subacute rehabilitation units from baseline to follow-up policy recommendations alignment, by implementation strategy group.**

**Table 6. Effect size estimates for primary and secondary outcomes using imputed data.**

| | Recommendation vs. control | Knowledge broker vs. control | Recommendation vs. knowledge broker | ICC* |
|---|---|---|---|---|
| **Primary** | | | | |
| Cluster level | | | | |
| Alignment with recommendations coefficient (95% CI) | 18.11 (−8,721.81 to 8,758.02) $p = 0.997$ | 1.24 (−6,992.60 to 6,995.07) $p = 1.000$ | −9.12 (−3,878.39 to 3,860.16) $p = 0.996$ | NA |
| Ward level | | | | |
| Alignment with recommendations OR (95% CI) | 0.01 (0.74 to 0.75) $p = 0.983$ | −0.12 (−0.54 to 0.30) $p = 0.581$ | −0.19 (−1.04 to 0.65) $p = 0.651$ | C: 0.31 H: 0.40 |
| **Secondary** | | | | |
| Ward level | | | | |
| Mean hospital length of stay coefficient (95% CI) | 2.19 (−1.36 to 5.74) $p = 0.219$ | −0.55 (−1.16 to 0.06) $p = 0.075$ | −3.75 (−8.33 to 0.82) $p = 0.102$ | C: 0.53 H: 0.56 |

*ICCs partitioned at the C and H levels.

C, cluster; H, hospital; ICC, intracluster correlation coefficient; NA, not applicable.

regarding weekend allied health service provision was the most common reason for nonparticipation. Two clusters (14%) did not engage with the knowledge broker without providing a reason. Further detail regarding the knowledge broker webinar support sessions is published elsewhere [58]. Other process, economic, and qualitative outcomes specified in our protocol are planned to be reported in other publications.

**Table 7. Complete case effect size estimates for primary and secondary outcomes sensitivity analysis.**

| | Recommendation vs. control | Knowledge broker vs. control | Recommendation vs. knowledge broker | ICC* |
|---|---|---|---|---|
| **Primary** | | | | |
| Cluster level | | | | |
| Alignment with recommendations coefficient (95% CI) | 17.29 (−5,508.16 to 5,542.76) $p = 0.995$ | 0.510 (−7,925.09 to 7,925.09) $p = 1.000$ | −16.68 (−7,035.87 to 7,002.51) $p = 0.996$ | NA |
| Ward level | | | | |
| Alignment with recommendations OR (95% CI) | 1.64 (0.62 to 4.33) $p = 0.315$ | 0.78 (0.48 to 1.28) $p = 0.330$ | 0.39 (0.14 to 1.11) $p = 0.078$ | C: 0.34 H: 0.38 |
| **Secondary** | | | | |
| Ward level | | | | |
| Mean hospital length of stay coefficient (95% CI) | 2.19 (−1.35 to 5.73) $p = 0.218$ | −0.55 (−1.16 to 0.06) $p = 0.074$ | −3.75 (−8.30 to 0.79) $p = 0.101$ | C: 0.36 H: 0.31 |

*ICCs partitioned at the C and H levels.

C, cluster; H, hospital; ICC, intracluster correlation coefficient; NA, not applicable.

## Discussion

This study was unable to identify differences between the knowledge broker and dissemination of written evidence-based practice recommendation strategies employed for hospital managers to improve the alignment of weekend allied health services with current evidence. These findings were possibly driven by the high rate of baseline alignment with the recommendations in the knowledge broker group and inadequate statistical power at the cluster-level analysis, despite including 132 hospitals and 833 wards. All 15 clusters assigned to the control group continued providing weekend allied health that was not aligned with the recommendations. Three clusters (21%) from the dissemination of written recommendations only group transitioned from nonalignment at baseline to partial alignment 12 months later. None of the clusters who received the knowledge broker in addition to the recommendations transitioned to partial or full alignment with the recommendations. The reduction in hospital length of stay for the knowledge broker group compared to the control indicates that there may have been an effect but the precise mechanism is unclear without an observed change in recommendation alignment.

Our study results are concordant with the only other three-arm trial published on the dissemination of recommendations and knowledge brokering for implementing evidence into healthcare policies and programs [36]. This earlier research was also unable to demonstrate a positive effect of their knowledge broker implementation strategy; however, the authors reported that impacts may have been more pronounced for health departments with low baseline organisational research culture. Achieving our desired outcomes at only a few organisations aligns with this, and other previous research indicating the effect of research implementation strategies may be contextually specific to organisational characteristics, such as strategic priority, leadership, and readiness [59,60]. The knowledge broker group in our study was characterised by higher baseline rates of recommendation alignment, despite pilot work conducted prior to study commencement indicating that most healthcare organisations were unlikely to be aligned with the evidence-based practice recommendations. While our analysis adjusted for baseline alignment, it is possible that a ceiling effect was reached in this group or other potentially confounding variables may have influenced differences in follow-up recommendation alignment. Formal assessment of evidence alignment and other potential confounders prior to study commencement was considered impractical within this study due to the data collection burden for time-limited hospital managers. Although, future research may benefit from these baseline assessments if information can be captured via routinely collected data for administrative or other purposes.

The decision to use an externally based, centralised knowledge broker to deliver support across multiple hospitals via interactive online webinar may have been an important factor influencing the strategy's impact. Our findings align with a similar cluster randomised controlled trial conducted by Minian and colleagues, which compared generic emails with a remote knowledge broker to integrate mood management into a smoking cessation program [61]. In their study, the more intense and personalised remote knowledge broker strategy was no more effective at enabling healthcare professionals to provide their patients with mood management resources. The "more is better" theory, suggesting that a higher implementation strategy dosage through frequency of interactions or longer duration leads to greater success, is difficult to reconcile with this emerging empirical evidence. Hospital managers and healthcare professionals are time and resource constrained, facing multiple competing priorities that limit their ability to engage in more active strategies to facilitate evidence implementation [41]. The limited frequency and duration of the knowledge broker interactions may have impeded observed effectiveness of the strategy. However, it is unlikely that providing additional

opportunities for engagement or a longer duration (e.g., 24 months) would have changed the dosage received, given the limited voluntary hospital manager engagement throughout the 12-month period and the desire to make an internal decision being the most common reason for limited engagement.

Most clusters only attended one or two group-based knowledge broker support sessions, which were delivered online rather than face-to-face. The limited dosage and mode of delivery may have constrained the ability to build relationships with the hospital managers. Knowledge broker roles are inherently relationship based [62], whose theory of change is premised on interpersonal contact [63], development of rapport [64], and building linkages and exchange between research producers and end users [31,39]. These social influence mechanisms of change [65–68] imply that the effectiveness of a knowledge broker may be largely individual dependent [53] and could be enhanced by embedded brokerage roles within organisations. For example, interactions between different hospital managers and knowledge brokers can result in varying levels of "relationship capital," which is instrumental to fostering use of knowledge [63]. Less formal roles that are internal to organisations, such as opinion leaders or clinical champions, could instead be considered to leverage preexisting, peer-to-peer relationships and channels of persuasion [63]. It is important for informal brokers to consider group affiliation with the hospital managers, through occupation and professional legitimacy, to ensure that brokerage is not considered as "outsider expertise" [38]. These localised models of knowledge brokering, such as the National Institute for Health Research (NIHR) Collaborations for Leadership in Applied Health Research and Care (CLAHRCs), may provide a more situationally relevant, flexible, and collaborative approach to implement research into practice [69], albeit at a higher cost of investment.

The small number of clusters in this study was primarily driven by the need to cluster multiple geographically distinct hospitals to avoid contamination. During protocol design and sample size calculation, it was not possible to anticipate how many wards would be included in each cluster prior to recruitment, as we first needed to ascertain the decision-making structure for each healthcare organisation to in order to understand how many wards were nested within each decision-making structure (e.g., single hospital or network of hospitals). We acknowledge that our efforts to understand and cluster according to these decision-making structures prevented prespecification of cluster size. This design element was imperative to avoid the potential risk of contamination between units of randomisation. Recruitment challenges and difficulty ensuring adequate statistical power in analyses have been reported in other studies seeking to implement evidence into health system policy changes [70], particularly for pragmatic "real-world" projects at the level of organisations rather than individuals [71,72]. Considering the challenges experienced in this study and others, some have questioned the appropriateness of empirical designs for evaluating the effectiveness of implementation strategies, such as knowledge brokering [36]. Instead, more exploratory approaches oriented towards improving the understanding of how, why, when, and in what circumstances these strategies address more subjective, intermediate outcomes (e.g., capability for evidence engagement) have been advocated. These types of pluralistic, discursive, and iterative methods hold considerable value for understanding the processes by which the institutionalisation of evidence-based practice can occur. However, the decision to delegate and allocate healthcare resources to specific individual knowledge brokers requires consideration of its effectiveness and cost-effectiveness, in relation to competing priorities that are known to effectively improve health outcomes elsewhere in the health system. We contend that there are design-based solutions for overcoming the challenges in empirically studying the effectiveness of implementation strategies. For example, the use of counterbalanced implementation study designs has been proposed as a solution to these challenges, requiring smaller sample sizes through the

concurrent investigation of multiple implementation strategies across different health context areas [73,74].

Weekend allied health service provision presented a uniquely challenging contextual area to implement evidence into practice. Healthcare professionals tend to incorporate local "tacit" or "codified" knowledge [75] into resource allocation decisions [76], particularly when presented with recommendations to reduce weekend allied health in acute general medical and surgical wards where the evidence base was considered less clear [58]. These findings align with previous research reporting that disinvestment from weekend services in these wards has been perceived as a threat to professional identity [77]. Conversely, recommendation to increase weekend services in subacute rehabilitation units was met with enthusiasm where this aligned with previously held attitudes, beliefs, and values [58]. Future research on these knowledge translation strategies is therefore needed across multiple contextual areas to determine the interaction effect between these strategies and context areas [73,74]. Further, an updated knowledge broker strategy with greater internal organisational relationships, which incorporates a component of cognitive debiasing to assist managers and policy makers when confronted with evidence that conflicts with current service delivery models, might reduce potential barriers to change and facilitate active implementation of evidence into practice [78–80].

## Conclusions

Owing to a lack of power at the cluster level, this trial was unable to determine whether the use of a webinar-based knowledge broker was more effective than dissemination of recommendations or usual practice for promoting evidence-informed healthcare management decisions for inpatient weekend allied health services. The implication of this research is that more intense and interactive strategies for implementing evidence into practice do not always enable changes in resource allocation. Future research is needed across multiple contextual areas of healthcare recommendations to understand the context dependency of implementation strategy success, using counterbalanced implementation study deigns.

## Supporting information

**S1 Data. Cluster-level per day values (blinded).**
(XLSX)

**S2 Data. Ward-level per day values (blinded).**
(XLSX)

**S1 Text. Study protocol.**
(PDF)

**S2 Text. CONSORT extension for cluster trials.**
(DOCX)

**S3 Text. Standards for Reporting Implementation Studies statement (StaRI).**
(DOCX)

**S4 Text. Weekend allied health recommendation.**
(PDF)

## Acknowledgments

The authorship team wish to acknowledge the contributions of Dr Jenni White and Dr Kellie Grant to the delivery of the implementation strategies and data collection within this study.

We would also like to recognise support from the Victorian Department of Health and Human services.

## Author Contributions

**Conceptualization:** Mitchell N. Sarkies, Cylie M. Williams, Nicholas F. Taylor, Lisa O'Brien, Jenny Martin, Anne Bardoel, Meg E. Morris, Leeanne M. Carey, Anne E. Holland, Terry P. Haines.

**Data curation:** Mitchell N. Sarkies, Lauren M. Robins, Megan Jepson, Terry P. Haines.

**Formal analysis:** Mitchell N. Sarkies.

**Funding acquisition:** Cylie M. Williams, Nicholas F. Taylor, Lisa O'Brien, Jenny Martin, Anne Bardoel, Meg E. Morris, Leeanne M. Carey, Anne E. Holland, Terry P. Haines.

**Investigation:** Mitchell N. Sarkies, Lauren M. Robins, Megan Jepson, Cylie M. Williams, Nicholas F. Taylor, Lisa O'Brien, Jenny Martin, Anne Bardoel, Meg E. Morris, Leeanne M. Carey, Anne E. Holland, Katrina M. Long, Terry P. Haines.

**Methodology:** Mitchell N. Sarkies, Cylie M. Williams, Nicholas F. Taylor, Lisa O'Brien, Jenny Martin, Anne Bardoel, Meg E. Morris, Leeanne M. Carey, Anne E. Holland, Katrina M. Long, Terry P. Haines.

**Project administration:** Mitchell N. Sarkies, Lauren M. Robins, Megan Jepson, Terry P. Haines.

**Supervision:** Mitchell N. Sarkies, Lauren M. Robins, Cylie M. Williams, Nicholas F. Taylor, Lisa O'Brien, Jenny Martin, Anne Bardoel, Meg E. Morris, Leeanne M. Carey, Anne E. Holland, Katrina M. Long, Terry P. Haines.

**Writing – original draft:** Mitchell N. Sarkies.

**Writing – review & editing:** Mitchell N. Sarkies, Lauren M. Robins, Megan Jepson, Cylie M. Williams, Nicholas F. Taylor, Lisa O'Brien, Jenny Martin, Anne Bardoel, Meg E. Morris, Leeanne M. Carey, Anne E. Holland, Katrina M. Long, Terry P. Haines.

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
