## [Editor Report · Decision Letter 0]

15 Mar 2021

Dear Dr Sarkies, 

Thank you for submitting your manuscript entitled "Effectiveness of knowledge brokering and recommendation dissemination on resource allocation decisions: cluster randomised controlled trial" for consideration by PLOS Medicine.

Your manuscript has now been evaluated by the PLOS Medicine editorial staff and I am writing to let you know that we would like to send your submission out for external peer review.

Please re-submit your manuscript within two working days, i.e. by March 18, 2021.

Kind regards,

Beryne Odeny

Associate Editor

PLOS Medicine

---

## [Decision Letter · Decision Letter 1]

6 May 2021

Dear Dr. Sarkies,

Thank you very much for submitting your manuscript "Effectiveness of knowledge brokering and recommendation dissemination on resource allocation decisions: cluster randomised controlled trial" (PMEDICINE-D-21-01199R1) for consideration at PLOS Medicine. 

[LINK]

Considering these reviews, we would be grateful if you could please revise your manuscript to respond to comments raised by reviewers. We would strongly recommend that you pay special attention to the statistical reviewers’ comments regarding your randomization procedures and would suggest toning down your conclusions. Please note that this is not a guarantee that we will accept the manuscript and that further consideration is dependent on the submission of a manuscript that addresses all reviewer concerns. We will carefully review your manuscript upon revision, so please ensure that your revision is thorough. 

Please also check the guidelines for revised papers at http://journals.plos.org/plosmedicine/s/revising-your-manuscript for any that apply to your paper. In your rebuttal letter you should indicate your response to the reviewers' and editors' comments, the changes you have made in the manuscript, and include either an excerpt of the revised text or the location (eg: page and line number) where each change can be found. Please submit a clean version of the paper as the main article file; a version with changes marked should be uploaded as a marked up manuscript.

We expect to receive your revised manuscript by May 27 2021 11:59PM. Please email us (plosmedicine@plos.org) if you have any questions or concerns.

We look forward to receiving your revised manuscript. 

Sincerely,

Beryne Odeny, 

PLOS Medicine

plosmedicine.org

- Please revise your title to indicate that this is an implementation science study. Your title must be nondeclarative and not a question. It should begin with main concept if possible. For example, please place the study design ("A cluster randomized controlled implementation trial,") in the subtitle (i.e., after a colon). 

- Abstract summary - At this stage, we ask that you reformat your non-technical Author Summary. The Author Summary should immediately follow the Abstract in your revised manuscript. This text is subject to editorial change and should be distinct from the scientific abstract. The summary should be accessible to a wide audience that includes both scientists and non-scientists. Please see our author guidelines for more information: https://journals.plos.org/plosmedicine/s/revising-your-manuscript#loc-author-summary.

- Abstract:

1. Please structure your abstract using the PLOS Medicine headings (Background, Methods and Findings, Conclusions).

2. Please combine the Methods and Findings sections into one section, “Methods and findings”. Please ensure that all numbers presented in the abstract are present and identical to numbers presented in the main manuscript text.

3. Please quantify the main results (with p values in addition to 95% CI).

4. Please include a summary of adverse events if these were assessed in the study.

- When completing the CONSORT checklist, please use section and paragraph numbers, rather than page numbers.

- In addition to the CONSORT checklist please ensure that your implementation research is reported according to Standards for Reporting Implementation Studies statement (STARI). The STARI guidelines can be found here: https://www.equator-network.org/reporting-guidelines/stari-statement/

- Please clearly specify and report your implementation strategies. Consider using the guidelines published by Proctor et al. to improve your reporting: Proctor EK, Powell BJ, McMillen JC: Implementation strategies: recommendations for specifying and reporting. Implement Sci 2013, 8:139 

- To improve the clarity and definition of your primary implementation outcomes, please use standard implementation science terminology such as adoption, fidelity, penetration, sustainability etc. 

- Please present p-values along with 95% CI in the tables and main text. Please specify the statistical test used to derive p values

- Please use the "Vancouver" style for reference formatting and see our website for other reference guidelines https://journals.plos.org/plosmedicine/s/submission-guidelines#loc-references. Please ensure that weblinks are current and accessible

Comments from the reviewers:

Reviewer #1: The evaluation of knowledge brokering is important to contribute to the knowledge translation literature. As the authors point out there have been relatively few empirical studies conducted to date evaluating the impact of knowledge brokering on the use of evidence-informed practices and/or recommendations. I enjoyed reading this article and believe, with revisions, it will be an important contribution to the literature. However, I have a few concerns that I feel are important to be addressed prior to publication. 

Major Concerns:

1) greater justification for the use of a cluster randomized controlled trial given previous suggestions in the literature to avoid the use of such a design to evaluate knowledge brokering.

2) More detail needs to be provided about the knowledge brokering intervention as this is very sparse.

3) However, my understanding from what is written about the knowledge brokering intervention is that in entailed interactive webinars. While this is certainly one activity that knowledge brokers engage in, if this was the only activity conducted during this study, I would not classify this as knowledge brokering. As the authors point out in the discussion, knowledge brokering involves building relationships with end users and considerable interaction. It does not appear from the description provided that these activities occurred, which might suggest that what was actually evaluated in this study was webinars as a knowledge translation strategy delivered by a knowledge broker. 

4) Also, previous research has suggested that knowledge brokering likely needs to be implemented for more than 12 months to be effective, so perhaps some more justification can be provided for why a 12 month intervention was implemented in this study. 

5) A significant challenge for this study is the extent to which it is under powered to detect even a large difference. I have two issues here. The sample size calculation is based on a very large effect (50% alignment with the allied health allocation recommendation). Justification from previous knowledge brokering studies for such a large effect should be provided. However, I am not certain that there is evidence to suggest that knowledge brokering evaluations in the past has found such large effects, in which case, a much more conservative effect size should have been used in the sample size calculation. The challenge this creates is that the required sample size would be much larger than it already is, which makes this study even more under-powered.

6) With the study being so significantly under-powered, the conclusion is worded much too strongly, that knowledge brokering is no more effective than dissemination of recommendations or standard practice. Given the lack of power to detect a difference, the only conclusion that can be drawn is that there was insufficient power to observe statistically significant differences between groups.

7) The ceiling effect in this study requires more attention in the discussion. While a regression analysis was conducted to take into account differences at baseline with half of the knowledge brokering group already being in alignment with the recommendations, this may not adequately adjust for these baseline differences. It is also worth noting that the knowledge brokering group had more highly educated participants than the other two groups, and could this explain the baseline difference in the primary outcome? 

8) I believe a much more developed discussion about the lessons learned about what went wrong in this trial would be extremely helpful in helping others fall in to the same challenges in evaluating knowledge brokering. This includes: appropriate designs for evaluation, a more fulsome knowledge brokering intervention implemented for a longer period of time, ensuring it is feasible to be adequately powered to detect statistically significant differences, and ensuring the outcome is not present prior to intervention. A comprehensive discussion of these issues, drawing from what is currently known in the literature, with suggestions on how to avoid these pitfalls, will provide an important contribution to the literature. 

Minor comments:

1) In the abstract it is stated the study occurred between Feb 2018 and Jan 2020, but in the Results section of the paper, it is stated data collection occurred until April 2020. 

2) Provide justification for why at the ward level, partial alignment with the recommendations was not included as an option for measuring the outcome.

3) It is stated in the methods that an intention to treat analysis was conducted. However, 4 organizations either did not provide any data or follow-up data and therefore were not included in the analysis. This is not consistent with intention to treat analysis. For an intention to treat analysis to have been conducted, those organizations that did not provide data should have still been analyzed in the group to which they were allocated. 

4) I may have missed it but I did not see statements as to there being statistically significant differences between groups at baseline. From the data provided it appears as though there could be. 

5) there is a formatting issue with Ref #30 

Reviewer #2: This is an interesting cluster randomised controlled trial on the effectiveness of knowledge brokering and recommendation dissemination on resource allocation decisions. However, the trial was poorly designed and conducted with quite a few major issues needing attention.

1) Cluster design. Throughout the paper, there is no detailed description of the cluster RCT design especially on cluster size. Also It's very confusing as to what was ultimately randomised? decision makers or wards?

2) Sample size calculation. On page 10 and 11, it said "a sample size of 25 clusters per group..." but in the results it said 15 clusters per group. Also, cluster size or ICC was never mentioned. And, why effect size is on cluster level rather than paticipants level? In short, the sample size calculation was very confusing and not informative therefore not adequate at all.

3) Primary outcome. It says on page 9 "The primary outcome of interest was whether weekend allied health service provision at both the cluster- and ward-level aligned with the recommendations at 12-months follow up". However, wards were not in the sample size calculation so not powered therefore should not be used as a primary outcome.

4) Intention to treat. It's claimed on page 10 "Analysis was undertaken according to intention-to-treat principles...". However, as shown in Figure 1 and table 1, the randomisation was disrupted at cluster level with error and empty clusters and also at healthcare decision maker level as shown in table 1 with huge imbalance. Basically, the trial analysis was not intention to treat at all, neither at cluster level nor paticipant level. The imbalance at both levels mean the trial was poor conducted and lost the purpose of effective randomisation. It basically becomes an observational study.

5) Statistical analysis. As the trial was not balanced at both levels, the simple summary statistics as shown in table 1-4 becomes inadequate and potentially misleading. Multivariable analysis adjusted for all confounders is essentially needed however it's not done or seen at all in results section although vaguely mentioned in the stats methods section. All these methodological inadequacies can lead to unreliable and unbelievable results and conclusions.

6) Randomisation. On page 10, not clear what was randomised? clusters or healthcare managers?

Reviewer #3: I wish to thank the editor for their invitation to review the manuscript, titled "Effectiveness of knowledge brokering and recommendation dissemination on resource allocation decisions: cluster randomised controlled trial." The aim of this study was to determine the effectiveness of two research implementation strategies to facilitate evidence-informed healthcare management decisions for the provision of inpatient weekend allied health services. Given that I am not a statistician and I would recommend the editors consider a statistical review prior to publication. I present my concerns in the order they appeared:

1. The Title: I find it a bit confusing and not specific enough 

2. The abstract: Might want to be specific as to which were the two implementation strategies used. 

3. The background section they might want to describe in more detail what is meant by "the dissemination of of evidence-based practice recommendations " (line 114-115), and the literature behind this strategy. 

4. How can the participants be blinded to receiving the support of a knowledge broker? 

5. How was the COM-B model used? 

6. Discussion: How do their finding relate to what previous researchers have found? 

7. What are the implications to the field? 

Reviewer #4: This study tried to address an important issue of the impact of evidence dissemination and knowledge brokering on implementation of health system evidence. It is a well designed cluster randomized trial, which is a strength. However there are a few significant shortcomings that affect the validity and usefulness of its conclusions.

-the interventions seem to be multi component and complex. The evidence-based recommendation states the desired objective but suggested complex activities to managers in order to achieve them. Activities such as development of a business case, identification of local priorities, engagement of diverse stakeholders including allied health professionals, nurses, and even patients in the process, and development of communication and negotiation channels with staff. It seems unlikely that email dissemination of an evidence-based guideline or one or two online webinars by the knowledge broker would be sufficient to change the behavior of hospital managers to engage in these complex and resource-intensive activities. 

-Since the authors did not report any information regarding the adoption and implementation of such activities (e.g how many managers initiated those activities, how many actually implemented them in their routine practice, how many were successful in engaging staff in planning and priority setting, etc), it is difficult to attribute the study findings to the lack of effectiveness or unsuccessful implementation. This is a widespread shortcoming in implementation studies, where the emphasis is on comparative effectiveness rather than investigating the conditions that facilitate or impede utilization and ultimate effectiveness of an implementation strategy. Ideally, a mixed methods study would provide better opportunities to address the contextual and procedural complexities of these interventions. But even as a proper RCT, a more comprehensive assessment of implementation outcomes was needed.

-ward-level analysis of alignment to the guidelines at study arms (table 4) shows that the number of wards with missing status increased dramatically at follow up in knowledge broker and dissemination arms. This along with the lack of engagement of sites in knowledge broker meetings implies that there were challenges in preserving the adherence of hospital mangers to the interventions. More clarification is needed regarding the potential reasons for this lack of adherence and the increase in non-response in intervention arms.

[LINK]

---

## [Decision Letter · Decision Letter 2]

14 Jul 2021

Dear Dr. Sarkies,

Thank you very much for submitting your manuscript "Effectiveness of knowledge brokering and recommendation dissemination for influencing healthcare resource allocation decisions: A cluster randomised controlled implementation trial" (PMEDICINE-D-21-01199R2) for consideration at PLOS Medicine. 

[LINK]

In light of these reviews, I am afraid that we will not be able to accept the manuscript for publication in the journal in its current form, but we would like to consider a revised version that addresses the reviewers' and editors' comments. Please pay particular attention to comments from reviewers #2 and #4. Obviously we cannot make any decision about publication until we have seen the revised manuscript and your response, and we plan to seek re-review by one or more of the reviewers. 

We expect to receive your revised manuscript by Aug 04 2021 11:59PM. Please email us (plosmedicine@plos.org) if you have any questions or concerns.

We look forward to receiving your revised manuscript. 

Sincerely,

Beryne Odeny, 

PLOS Medicine

plosmedicine.org

Comments from the reviewers:

Reviewer #1: The authors have comprehensively addressed all of my suggestions. 

Reviewer #2: Many thanks authors for their great effort to improve the manuscript. I can appreciate the challenges in this implementation trial that investigators have faced and the fact that investigators tried their best to deal with the situations, however, I am still not convinced/statisfied with the response and revision. Remaing issues:

1) This is not an "intention to treat" analysis. There are 3 clusters without 12-month follow up (primary) so not in the main analysis, as shown in Figure 1. Also multiple imputation for missing data is not ITT analysis. So far, it was effectively performed with a complete case analysis plus sensitivity analysis for missing values (imputation).

2) Design issue. Cluster RCT without pre-specified or uncertain cluster size? Sample size only for cluster level but not for ward level? Primary outcome for both cluster (powered) and ward (not powered)? All these deviate from the definition of a cluster RCT. Perhaps this is the pragmatic nature of implementation trial but how to address the methodological issues and potential biases which could impact on the results?

3) Statistical analysis. Again, the trial was not balanced at both levels, the simple summary statistics becomes inadequate and potentially misleading. Multivariable analysis comprehensively adjusted for all confounders is essentially needed. Otherwise, the results may be subject to scrutiny due to design, sample size and trial conduct issues. 

Reviewer #3: The authors have addressed all the comments in a satisfactory manner.

Reviewer #4: The authors tried to address most of the comments by reviewers. The revised manuscript is better organized and is more informative.

My main remaining concern, which is now more prominent since the term 'implementation' was added to the title, is lack of a guiding framework and evaluation metrics related to the implementation. Alignment of practice with the guidelines does not capture the complexities of the implementation. It seems that some data have already been collected that should be added to the Results. 

-For example webinar attendance rates at ward and organization levels may address the notion of 'adoption' of the intervention by managers. Furthermore, the authors can include ward-level attendance data in the regression analyses to assess its impact on study outcomes.

-Also, given the large non-response, it is important to provide some contextual information about the characteristics of non-respondents, in comparison to the participating organizations. This comparison may reflect the notion of 'reach' (who was missed?), using RE-AIM terminology.

-Providing some temporal participation and adoption statistics (how did the participation differ during the first few months vs later dates), if available, would address the notion of sustainability ('maintenance' in RE-AIM).

[LINK]

---

## [Decision Letter · Decision Letter 3]

13 Sep 2021

Dear Dr. Sarkies,

Thank you very much for re-submitting your manuscript "Effectiveness of knowledge brokering and recommendation dissemination for influencing healthcare resource allocation decisions: A cluster randomised controlled implementation trial" (PMEDICINE-D-21-01199R3) for review by PLOS Medicine.

I have discussed the paper with my colleagues and the academic editor and it was also seen again by two reviewers. I am pleased to say that provided the remaining editorial and production issues are dealt with we are planning to accept the paper for publication in the journal.

[LINK]

We look forward to receiving the revised manuscript by Sep 20 2021 11:59PM.   

Sincerely,

Beryne Odeny, 

Associate Editor 

PLOS Medicine

plosmedicine.org

Requests from Editors:

Thank you for responding to editorial requests. Before we proceed, please address the following comments:

1) Please highlight in the conclusion (both abstract and main text), that you did not observe a difference imparted by the implementation strategies, owing to lack of power. For example, you could state “owing to lack of power, our study was unable to find whether…”

2) In the author summary, please revise line 110... should this be “are” instead of “are represent”?

3) In the abstract, last sentence of “Methods and Findings” please clearly highlight the limitations of the study. The statement, “Limitations of this study include…” can be useful.

4) References #55, #57: please include access dates for the weblinks, e.g., Accessed July 15, 2021. 

Comments from Reviewers:

Reviewer #2: The authors have addressed all my concerns professionally. I am satisfied with the response and revision. No further issues needing attention. Thanks.

Reviewer #4: I'd like to thank the authors for their efforts to address the comments.

I think that the researchers could use more suitable implementation theoretical frameworks to guide the design of the study, as well as more relevant and sensitive implementation outcomes to capture the impact of individual, team-level, and organizational factors that could determine the success/failure of the intervention. Despite this, I do not object the publication of this study, given its rigorous design and potentially important implications for further research on knowledge broker interventions. 

However, I re-emphasize that, it should be clearly stated in the study limitations that the observed lack of evidence to support knowledge broker intervention could simply be due the fact that the intervention was not successfully implemented (for which insufficient evidence was provided). I recommend that the authors suggest potential solutions for more theoretically-informed assessment of the implementation of KB interventions.

[LINK]

---

## [Editor Report · Decision Letter 4]

4 Oct 2021

Dear Dr Sarkies, 

On behalf of my colleagues and the Academic Editor, Dr. Elvin Hsing Geng, I am pleased to inform you that we have agreed to publish your manuscript "Effectiveness of knowledge brokering and recommendation dissemination for influencing healthcare resource allocation decisions: A cluster randomised controlled implementation trial" (PMEDICINE-D-21-01199R4) in PLOS Medicine.

PRESS

Sincerely, 

Beryne Odeny 

Associate Editor 

PLOS Medicine